Extended Abstract Track

# Towards Architectural Optimization of Equivariant Neural Networks over Subgroups

**Kaitlin Maile**                                             KAITLIN.MAILE@IRIT.FR
*IRIT, University of Toulouse*

**Dennis G. Wilson**                              DENNIS.WILSON@ISAE-SUPAERO.FR
*ISAE-SUPAERO, University of Toulouse*

**Patrick Forré**                                               P.D.FORRE@UVA.NL
*University of Amsterdam*

**Editors:** Sophia Sanborn, Christian Shewmake, Simone Azeglio, Arianna Di Bernardo, Nina Miolane

## Abstract

Incorporating equivariance to symmetry groups in artificial neural networks (ANNs) can improve performance on tasks exhibiting those symmetries, but such symmetries are often only approximate and not explicitly known. This motivates algorithmically optimizing the architectural constraints imposed by equivariance. We propose the equivariance relaxation morphism, which preserves functionality while reparameterizing a group equivariant layer to operate with equivariance constraints on a subgroup, and the $[G]$-mixed equivariant layer, which mixes operations constrained to equivariance to different groups to enable within-layer equivariance optimization. These two architectural tools can be used within neural architecture search (NAS) algorithms for equivariance-aware architectural optimization.

**Keywords:** equivariance, neural architecture search, geometric deep learning

## 1. Introduction

Incorporating constraints of symmetry group equivariance into neural networks can improve their task performance, efficiency, and generalization capabilities (Bronstein et al., 2021), as shown by translation-equivariant convolutional neural networks for image-based tasks (Fukushima and Miyake, 1982; LeCun et al., 1989). Seminal works have developed general theories and architectures for equivariance in neural networks, providing a blueprint for equivariant operations on complex structured data (Cohen and Welling, 2016; Ravanbakhsh et al., 2017; Kondor and Trivedi, 2018; Weiler et al., 2021). However, these works design model constraints based on an explicit equivariance property. Furthermore, the architectural assumption of full equivariance may be overly constraining; e.g., in handwritten digit recognition, full equivariance to 180° rotation may lead to misclassifying samples of "6" and "9". Weiler and Cesa (2019) found that local equivariance from a final subgroup convolutional layer improves performance over full equivariance. If appropriate equivariance constraints are instead learned, the benefits of equivariance could extend to applications where the data may have unknown or imperfect symmetries.

Learning approximate equivariance has lately been approached through novel layer operations. Wang et al. (2022) relaxes equivariance via low-rank partial expansion. Finzi et al.

# Extended Abstract Track

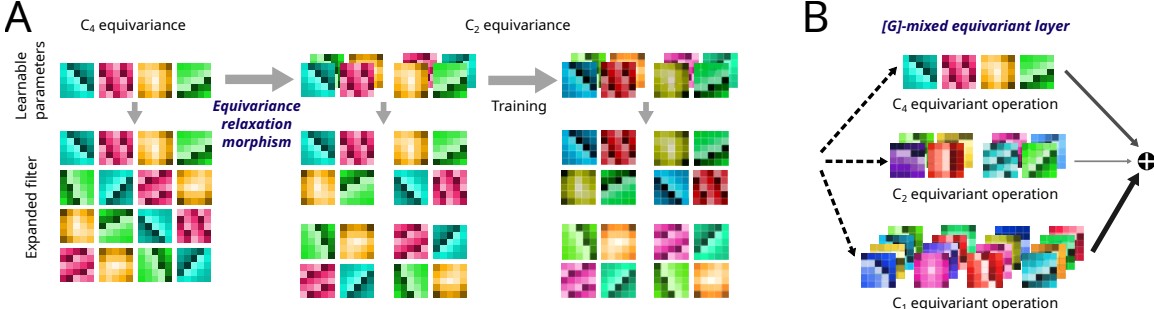

Figure 1: Visualizations of (A) the equivariance relaxation morphism and (B) the $[G]$-mixed equivariant layer, using the $C_4$ group. In (A), the learnable parameters of a $C_4$-equivariant convolutional layer are expanded using the group element actions, such that the expanded filter can be used in a standard convolutional layer. Applying the equivariance relaxation morphism reparameterizes the layer to only be architecturally constrained to $C_2$ equivariance, initialized to be functionally $C_4$ equivariant. In (B), convolutional operations equivariant to subgroups of $C_4$ are summed with learnable architectural weighting parameters.

(2021) uses an equivariant layer in parallel to a non-equivariant layer with weighted regularization. Zhou et al. (2020) and Yeh et al. (2022) represent symmetry-inducing weight sharing through learnable matrices. Separately, the field of Neural Architecture Search (NAS) aims to optimize full neural network architectures. Existing NAS methods have not yet been developed for optimizing equivariance, although Romero and Lohit (2022) and van der Ouderaa et al. (2022) learn partial or soft equivariances per layer towards custom equivariant architectures. An important aspect of NAS is network morphisms: function-preserving architectural changes (Wei et al., 2016) used during training to change the loss landscape and gradient descent trajectory while immediately maintaining the current functionality and loss value (Maile et al., 2022). Agrawal and Ostrowski (2022) present theoretical contributions on subgroup-based network morphisms for group-invariant shallow neural networks. Our work focuses on layer-wise equivariance for deep convolutional architectures. Developing tools for searching over architectural representations of equivariance permits the application of NAS algorithms towards architectural optimization of equivariance.

We present two contributions aimed to enable the search for appropriate equivariance in the following section. The first is the *equivariance relaxation morphism* for group convolutional layers that partially expands the parameters of the layer to enable less constrained learning with a prior on symmetry. Secondly, we define the $[G]$-*mixed equivariant layer* that parameterizes a layer as a weighted sum of layers equivariant to different groups, allowing for the learning of architectural weighting parameters. We conclude by proposing extensions of these theoretical results in equivariance-aware neural architecture search in Appendix A, showing experimental results in Appendix B.

Extended Abstract Track

## 2. Towards Architectural Optimization over Subgroups

We propose two mechanisms to enable search over subgroups: the equivariance relaxation morphism and a $[G]$-mixed equivariant layer, visualized in Figure 1. The proposed morphism changes the equivariance constraint from one group to another subgroup while preserving the learned weights of the initial group convolutional operator; this is useful for architecture search methods like evolutionary NAS that navigate between discrete architectural choices (Elsken et al., 2017; Lu et al., 2019). The second contribution, the $[G]$-mixed equivariant layer, allows for a single layer to represent equivariance to multiple subgroups through a weighted sum. Learning the weights of such a hypernetwork is common in differentiable NAS approaches, such as DARTS (Liu et al., 2018). We first present a preliminary background on group convolutions and then the theoretical foundations of the two proposed mechanisms.

### 2.1. Group Convolution Background

We assume familiarity with group theory (Herstein, 2006). Let $G$ be a discrete group. The $l^{\text{th}}$ $G$-equivariant group convolutional layer (Cohen and Welling, 2016) of a group convolutional neural network (G-CNN) convolves the feature map $f\colon G \to \mathbb{R}^{C_{l-1}}$ output from the previous layer with a filter with kernel size $k$ represented as learnable parameters $\psi\colon G \to \mathbb{R}^{C_l \times C_{l-1}}$. For each output channel $d \in [C_l]$ where $[C] := \{1, \ldots, C\}$, and group element $g \in G$, the layer's output is defined via the convolution operator[1]:

$$[f \star_G \psi]_d(g) = \sum_{h \in G} \sum_{c=1}^{C_{l-1}} f_c(h)\psi_{d,c}(g^{-1}h). \tag{1}$$

The first layer is a special case: in the case of image data, the input is $x\colon \mathbb{Z}^2 \to \mathbb{R}^{C_0}$, so the layer instead performs a *lifting* convolution:

$$[x \star_G \psi]_d(g) = \sum_{y \in \mathbb{Z}^2} \sum_{c=1}^{C_0} x_c(y)\psi_{d,c}(g^{-1}y). \tag{2}$$

We present our contributions in the group convolutional layer case, although similar claims apply for the lifting convolutional layer case.

### 2.2. Equivariance Relaxation Morphism

The *equivariance relaxation morphism* reparameterizes a $G$-equivariant group (or lifting) convolutional layer to operate over a subgroup of $G$, partially removing weight-sharing constraints from the parameter space while maintaining the functionality of the layer, shown in Figure 1(A).

Let $G' \leq G$ be a subgroup of $G$. Let $R$ be a system of representatives of the left quotient (including the neutral element), so that $G'\backslash G = \{G'r \mid r \in R\}$, where $G'r := \{g'r \mid g' \in G'\}$. Given a $G$-equivariant group convolutional layer with feature map $f$ and filter $\psi$, we define

---

1. We identify the correlation and convolution operators as they only differ where the inverse group element is placed and refer to both as "convolution" throughout this work.

the relaxed feature map $\tilde{f} \colon G' \to \mathbb{R}^{(C_{l-1} \times |R|)}$ and relaxed filter $\tilde{\psi} \colon G' \to \mathbb{R}^{(C_l \times |R|) \times (C_{l-1} \times |R|)}$ as follows. For $c \in [C_{l-1}]$, $s, t \in R$, $d \in [C_l]$:

$$\tilde{f}_{(c,s)}(g') := f_c(g's), \tag{3}$$

$$\tilde{\psi}_{(d,t),(c,s)}(g') := \psi_{d,c}(t^{-1}g's). \tag{4}$$

We define the *equivariance relaxation morphism* from $G$ to $G'$ as the reparameterization of $\psi$ as $\tilde{\psi}$ (Eq. 4) and reshaping of $f$ as $\tilde{f}$ (Eq. 3). We will show that the new output, $[\tilde{f} \star_{G'} \tilde{\psi}]_{(d,t)}(g')$, is equivalent to $[f \star_G \psi]_d(g't)$ down to reshaping. Since the mapping $G' \times R \to G$, $(g', t) \mapsto g't$, is bijective, every $g$ can uniquely be written as $g = g't$ with $g' \in G'$ and $t \in R$. For $g \in G$, $G'g \in G' \setminus G$ has a unique representative $t \in R$ with $G'g = G't$, and $g' := gt^{-1} \in G'$. By the same argument, $h \in G$ may be written as $h = h's$ with unique $h \in G'$ and $s \in R$. With these preliminaries, we get:

$$[f \star_G \psi]_d(g't) = [f \star_G \psi]_d(g) \tag{5}$$

$$= \sum_{h \in G} \sum_{c=1}^{C_{l-1}} f_c(h)\psi_{d,c}(g^{-1}h), \tag{6}$$

$$= \sum_{h' \in G'} \sum_{s \in R} \sum_{c=1}^{C_{l-1}} f_c(h's)\psi_{d,c}(t^{-1}g'^{-1}h's), \tag{7}$$

$$= \sum_{h' \in G'} \sum_{c=1}^{C_{l-1}} \sum_{s \in R} \tilde{f}_{(c,s)}(h')\tilde{\psi}_{(d,t),(c,s)}(g'^{-1}h'), \tag{8}$$

$$= \left[ \tilde{f} \star_{G'} \tilde{\psi} \right]_{(d,t)}(g'), \tag{9}$$

which shows the claim. Thus, the convolution of $\tilde{f}$ with $\tilde{\psi}$ is equivariant to $G$ but parametrized as a $G'$-equivariant group convolutional layer, where the representatives are expanded into independent channels. This morphism can be viewed as initializing a $G'$-equivariant layer with a pre-trained prior of equivariance to $G$, maintaining any previous training.

Standard convolutional layers are a special case of group-equivariant layers, where the group is translational symmetry over pixel space. Regular group convolutions are often implemented by relaxation to the translational symmetry group by expanding the filter via the appropriate group actions, allowing a standard convolution implementation from a deep learning library to be used. The equivariance relaxation morphism generalizes this concept to any subgroup.

## 2.3. $[G]$-Mixed Equivariant Layer

Towards learning equivariance, we additionally propose partial equivariance through a mixture of layers, each constrained to equivariance to different groups and applied in parallel to the same input then all combined via a weighted sum, shown in Figure 1(B). The equivariance relaxation morphism provides a mapping of group elements between pairs of groups where one is a subgroup of the other. For a set of groups $[G]$ where each group is a subgroup

# Extended Abstract Track

or supergroup of all other groups within the set, we define a $[G]$-*mixed equivariant layer* as:

$$\left[f \hat{\star}_{[G]}[\psi]\right]_{(d,t)}(g) = \sum_{G \in [G]} z_G \left[f \star_{G'} \widetilde{\psi^G}\right]_{(d,t)}(g) \tag{10}$$

$$= \left[f \star_{G'} \sum_{G \in [G]} z_G \widetilde{\psi^G}\right]_{(d,t)}(g), \tag{11}$$

where each element $z_G$ of $[z] \coloneqq \{z_G | G \in [G]\}$ is an architectural weighting parameter such that $\sum_{G \in [G]} z_G = 1$, $G'$ is a subgroup of all groups in $[G]$, each element $\psi^G$ of $[\psi]$ is a filter with a domain of $G$, and $\widetilde{\psi^G}$ is the transformation of $\psi^G$ from a domain of $G$ to $G'$ as defined in Equation 4. Thus, the layer is parametrized by $[\psi]$ and $[z]$, computing a weighted sum of operations that are equivariant to different groups of $[G]$. The layer may be equivalently computed by convolution of the input with the weighted sum of transformed filters, shown in Equation 11.

## 3. Conclusion

Towards architectural optimization of partial equivariance based on subgroup decomposition, we propose the equivariance relaxation morphism and the $[G]$-mixed equivariant layer. The equivariance relaxation morphism can be utilized in a simple evolutionary NAS algorithm, while the $[G]$-mixed equivariant layer can be optimized with a differentiable NAS algorithm. Such algorithms are proposed in Appendix A, with experimental results following in Appendix B. For the $[G]$-mixed equivariant layer, architectural discretization and parameter retraining are not necessary as in other differentiable NAS algorithms, as the mixture of filters can be combined into a single filter (Eq. 11). As such, the $[G]$-mixed equivariant layer could be directly used as a layer in standard network training, beyond NAS algorithms; the equivariance relaxation morphism could also be extended to applications beyond architecture search such as fine-tuning and distillation. By demonstrating that groups can be relaxed or mixed in a single layer, we aim to facilitate the learning of networks which automatically find optimal partial equivariance.

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

## Appendix A. Equivariance-Aware Neural Architecture Algorithms

We present two neural architecture search methods that utilize the presented mechanisms for discovering appropriate equivariance during neural network training: Evolutionary Equivariance-Aware NAS (EquiNAS$_E$) and Differentiable Equivariance-Aware NAS (EquiNAS$_D$). Both methods optimizffe an architecture while learning network weights, returning a final trained network adapted to the equivariances present in the training data. However, they differ in NAS paradigm and approximate equivariance representation: EqufiNAS$_E$, described in Section A.1, searches for networks composed of layers each fully equivariant to possibly different groups, while EquiNAS$_D$, described in Section A.2, searches for smooth mixtures of equivariant layers.

### A.1. Evolutionary Equivariance-aware NAS

Towards finding the optimal full equivariance per layer, the equivariance relaxation morphism presented in Section 2.2 is applied as the genetic operator in an evolutionary hill-climbing algorithm. The Evolutionary Equivariance-Aware NAS (EquiNAS$_E$) algorithm, given in Algorithm 1, is similar to other evolutionary NAS methods such as Elsken et al. (2017) with pareto selection as in Falanti et al. (2022). A population of networks, which starts with an individual with all layers equivariant to the largest possible group, undergoes mutation via equivariance relaxation and selection based on accuracy and parameter count to optimize neural architecture while learning network parameters.

In each generation, candidate networks are evaluated based on maximizing validation accuracy and minimizing parameter count: the entire Pareto front is kept, then additional high-accuracy individuals are added if necessary until the desired parent population size is reached. Offspring are generated from each parent separately through mutation using

# Extended Abstract Track

---

**Algorithm 1:** EquiNAS$_E$: Evolutionary equivariance-aware neural architecture search.

---

**Input:** Initial symmetry group $G$.

**Output:** Population of trained networks.

Initialize population with a $G$-equivariant group convolutional network.

**for** *each generation* **do**

    **for** *each network in population* **do**

       | Add children of network with relaxed equivariance constraints into population.

    **end**

    **for** *each network in population* **do**

       | Partially train network on dataset.

    **end**

    Select Pareto-efficient and high accuracy networks as new population.

**end**

---

the relaxation morphism. This preserves the weights of the parameterized equivariance during mutation, allowing for the continuous training of networks over evolution through inheritance from parent individuals. Specifically, mutation reduces a single layer's parameterized equivariance to a subgroup within the constraint that each layer has parametrized equivariance to a subgroup of all preceding layers. This constraint yields local equivariance properties for the network, as shown in Weiler and Cesa (2019) and Elsayed et al. (2020) to be empirically favorable in image classification tasks. The resulting individuals are each trained independently for a given training time, and then this process repeats.

The second objective of minimizing parameter count is intended to advance efficient networks, such as those with large symmetry groups. Accuracy-based selection alone would necessarily prefer larger networks as mutation via the equivariance relaxation morphism results in two networks with identical performance but different size, the relaxed network having more parameters, until training; potentially short-term increases in validation accuracy after training would then result in the selection of individuals with more parameters. Thus, the proposed strategy of selecting both pareto-front and high-accuracy individuals is intended to maintain a diverse yet efficient population without succumbing to overly greedy selections too early.

### A.2. Differentiable Equivariance-aware NAS

In a contrasting paradigm, the $[G]$-mixed equivariant layer presented in Section 2.3 allows for smoothly searching across a spectrum of equivariance for each layer via a differentiable NAS algorithm. Our Differentiable Equivariance-Aware NAS (EquiNAS$_D$) algorithm, defined in Algorithm 2, is inspired by DARTS (Liu et al., 2018) with significant changes detailed in the following paragraphs. EquiNAS$_D$ simplifies the bilevel optimization of the architecture weighting parameters $Z$ and filter weights $\Psi$ into alternating independent updates, computing the gradient update for $Z$ with the current, rather than optimal, $\Psi$ for the current architecture encoded by $Z$, to boost search efficiency with minimal performance loss compared to higher order approximations (Liu et al., 2018).

# Extended Abstract Track

---

**Algorithm 2:** EquiNAS$_D$: Differentiable equivariance-aware neural architecture search.

---

**Input:** Set of groups $[G]$.
**Output:** Trained network.
Initialize network with $[G]$-mixed equivariant layers, parameterized by $\Psi$ and $Z$.
**while** *not converged* **do**
    | Update $Z$ by $\nabla_Z \mathcal{L}(\Psi, Z)$.
    | Update $\Psi$ by $\nabla_\Psi \mathcal{L}(\Psi, Z)$.
**end**

---

In most differentiable NAS search spaces, the desired output architecture is discretized to select a subset of architectural options within constraints, then the weights are re-initialized and trained within the static architecture. In our formulation, this is not necessary as any mixed operation can be equivalently expressed as a single layer equivariant to any group $G'$ that is a common subgroup to all groups of the mixed operation (Eq. 11): in our experimental case, this is a standard translation-equivariant convolutional layer, so the final model can be equivalently expressed as a standard convolutional model with encoded partial equivariance. Thus, the final optimized architecture and trained weights are output from the single search process.

In order to enforce that the scaling of each filter does not confound the architecture weighting parameters, we use the weight normalizing reparameterization (Salimans and Kingma, 2016) and do not update the scalar norm parameter of each filter after initialization.

We do not use disjoint datasets for updating $\Psi$ and $Z$, but rather draw one batch for $\Psi$ and another for $Z$ independently and randomly from the same training split. This allows for a standard dataset split and to use the validation set for hyperparameter tuning.

These two NAS approaches present adaptations of two standard types of NAS, evolutionary and differentiable, to the search for optimal partial equivariance. The equivariance relaxation morphism and the $[G]$-mixed equivariant layer that enable the evolutionary and differentiable search methods respectively are the main focus; the other characteristics of the NAS methods are adapted from existing methods, but further study could advance specialization of equivariance-aware NAS. We next study empirically the two EquiNAS methods on three datasets, one with known rotational symmetry and two with unknown but visually significant rotational and reflectional symmetry.

## Appendix B. Results

We focus on the regular representation of groups and show experiments with reflectional and up to 4-fold rotational symmetry groups applied to image classification tasks. Examples of symmetry groups acting on pixel space, which corresponds to $\mathbb{Z}^2$, include $T(2)$, which consists of discrete translations in both dimensions; the cyclical groups $C_n$, which consist of $n$-fold rotations; and the dihedral groups $D_n$, which consist of reflections with $n$-fold rotations, where $n \in \{1, 2, 4\}$ for exact symmetry without interpolation. The $p4$ group consists of discrete translations and multiples of $90°$ rotations and may be represented as $T(2) \rtimes C_4$. The $p4m$ group consists of discrete translations, reflections, and multiples of $90°$

rotations and may be represented as $T(2) \rtimes D_4$. As standard convolutional layers are already equivariant to $T(2)$, we refer to layers also equivariant to $n$-fold rotations with or without reflections as $D_n$ or $C_n$-equivariant, respectively. So, a $C_1$ equivariant convolutional layer is a standard translation-equivariant convolutional layer. We use $\{C_1, D_1, C_2, D_2, C_4, D_4\}$ as the set of potential groups for mutation in EquiNAS$_E$ and as $[G]$ in EquiNAS$_D$.

We present experiments on image classification for a variety of datasets. The Rotated MNIST dataset (Larochelle et al., 2007, rotMNIST) is a version of the MNIST handwritten digit dataset but with the images rotated by any angle. This task serves as a simple investigational study with known symmetry, while the following two tasks are more realistic and complex. The Galaxy10 DECals dataset (Leung and Bovy, 2019, Galaxy10) contains galaxy images in 10 broad categories. The ISIC 2019 dataset (Codella et al., 2018; Tschandl et al., 2018; Combalia et al., 2019, ISIC) contains dermascopic images of 8 types of skin cancer plus a null class. For Galaxy10 and ISIC, we down-sample the images to $64 \times 64$ due to computational constraints, which adds notable difficulty to the tasks. These tasks exhibit varying levels of rotational and reflectional symmetry, motivating architectural optimization to determine the most effective application of equivariance constraints.

Across all experiments, the architectures are designed to have consistent channel dimensions once expanded to a standard translation-equivariant convolutional layer for each layer across models. Thus, constrained equivariance to a larger symmetry group results in fewer learnable parameters. A layer constrained to $C_4$ equivariance has $|C_4 \setminus D_4| = 2$ times as many independent channels and as many parameters as a layer constrained to $D_4$ equivariance. This is a notably different paradigm than other works that equate parameter counts across architectures with different equivariance properties.

As baseline comparisons, we train and test G-CNNs with static architectures. In addition to the static baselines, we re-implement the residual pathway priors (RPP) approach by Finzi et al. (2021) as a $C_1$ equivariant layer with regularization in parallel with a $D_4$ equivariant convolutional layer.

Further experiment details such as architecture details and other hyperparameters can be found in Appendix D. For each paradigm of experiments, we present their results in the following subsections, with general discussion following in Section B.3.

### B.1. Evolutionary Equivariance-aware NAS

The classification test errors are listed in Table 1. The advantages of equivariance search methods are most apparent in the Galaxy10 benchmark. While EquiNAS$_E$ outperforms most baselines on rotMNIST and all baselines on ISIC, it has similar performance on both tasks to the $D_4$ baseline, and some of the final architectures are very similar to the $D_4$ baseline architecture. However, the $D_4$ baseline fails at the Galaxy10 task, demonstrating that the same equivariant architecture can not be naively applied to different tasks. Both search methods, EquiNAS$_E$ and RPP, outperform all baseline models on Galaxy10, and by a large margin for EquiNAS$_E$.

The evolutionary progress on rotMNIST is shown in Figure 2: the selected population maintains a fully equivariant network in every generation. The final selected population originates from two main lineages, one staying fully equivariant until the last generations

| Method | rotMNIST | Galaxy10 | ISIC |
|---|---|---|---|
| EquiNAS$_E$ | **1.78 ± 0.04** | **__20.3 ± 0.9__** | **__31.0 ± 0.4__** |
| RPP (Finzi et al., 2021) | 2.18 ± 0.04 | **24.3 ± 2.8** | 32.2 ± 1.7 |
| $D_4$ baseline | **1.78 ± 0.11** | 50.8 ± 17.0 | 32.1 ± 2.4 |
| $C_4$ baseline | **__1.64 ± 0.22__** | 29.6 ± 5.5 | 32.9 ± 1.0 |
| $C_1$ baseline | 5.02 ± 1.15 | 31.6 ± 4.8 | 33.2 ± 1.5 |
| $C_4$ (prior: $D_4$) | 1.93 ± 0.05 | 27.8 ± 5.3 | 31.9 ± 1.5 |
| $C_1$ (prior: $D_4$) | 3.40 ± 0.07 | 25.9 ± 2.3 | **31.4 ± 2.6** |
| $C_1$ (prior: $C_4$) | 2.96 ± 0.05 | 30.7 ± 7.1 | 32.5 ± 1.1 |

Table 1: Test error percentages (lower is better) across tasks and approaches. Statistics are aggregated over the final selected population of 5 individuals for EquiNAS$_E$ and across 5 random seeds for all other methods. The **__best__** and **second best** average errors for each task are highlighted. See Figure 4 in Appendix E for individual trial results.

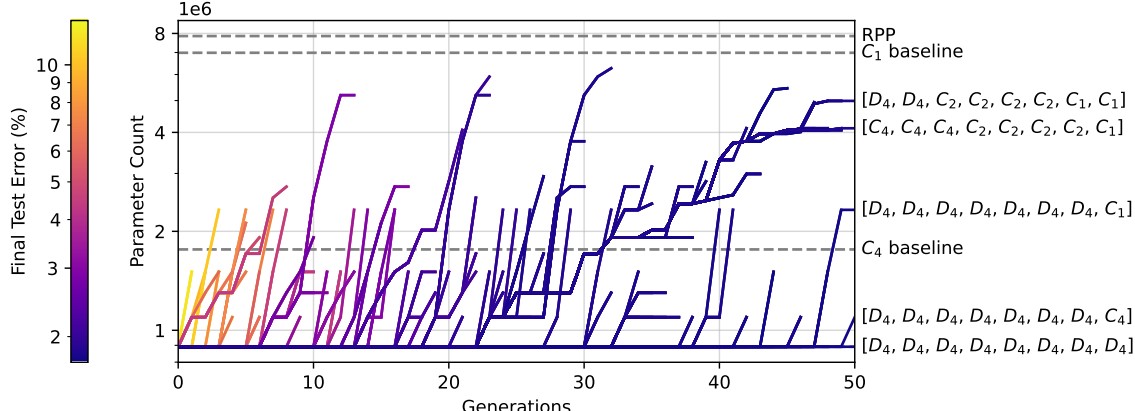

Figure 2: Historical parameter counts of each selected individual for EquiNAS$_E$ on rotM-NIST. The architectures of the final selected population are labeled. Each parameter count history is colored according to the final test accuracy, which is measured of each individual upon removal. For other tasks, see Figures E-E in Appendix E.

and the other diverging from the fully equivariant network midway through, showing that training with dynamically constrained parameterizations can produce performant models.

In addition to the normally initialized static baselines, we also train and test baselines that are initialized with priors to larger symmetry groups. These are implemented by initializing all layers to be constrained to the prior symmetry group, then using the equivariance relaxation morphism on each layer. EquiNAS$_E$ searches for relaxation schedules that yield trained priors on equivariance, while these additional baselines yield untrained priors. The

| Method | rotMNIST | Galaxy10 | ISIC |
|---|---|---|---|
| EquiNAS$_D$ | $\underline{\mathbf{2.29 \pm 0.27}}$ | $\underline{\mathbf{21.8 \pm 1.2}}$ | $32.8 \pm 0.6$ |
| RPP (Finzi et al., 2021) | $2.89 \pm 0.27$ | $\mathbf{22.0 \pm 1.8}$ | $\underline{\mathbf{31.5 \pm 0.9}}$ |
| $D_4$ Baseline | $2.97 \pm 1.50$ | $22.5 \pm 2.0$ | $\mathbf{32.0 \pm 1.0}$ |
| $C_4$ Baseline | $\mathbf{2.43 \pm 0.54}$ | $22.2 \pm 2.4$ | $32.8 \pm 1.0$ |
| $C_1$ Baseline | $3.97 \pm 0.75$ | $26.5 \pm 1.5$ | $32.9 \pm 3.1$ |

Table 2: Test error percentages (lower is better) in percent of incorrect classifications across tasks and approaches. The **best** and **second best** average errors for each task are highlighted. See Figure 7 in Appendix E for individual trial results.

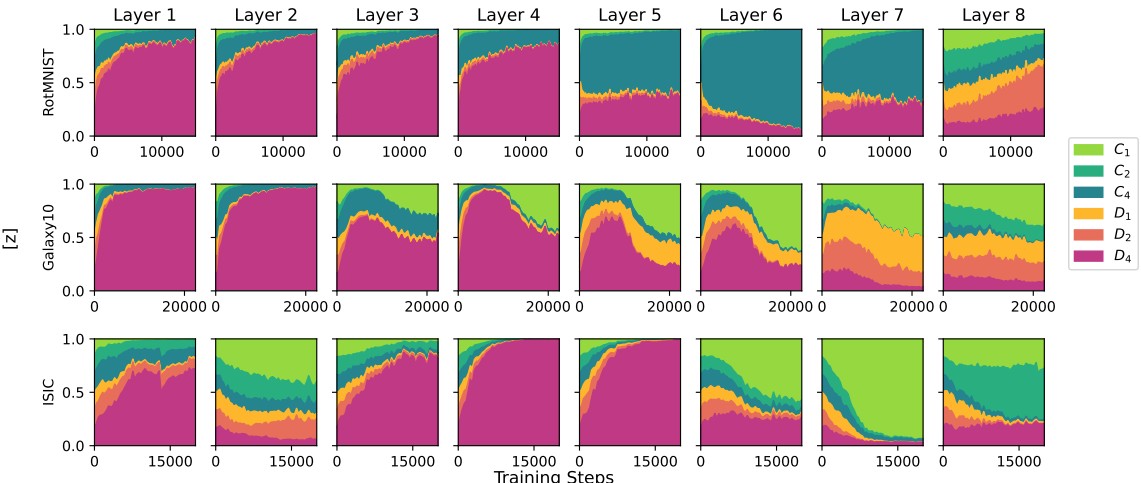

Figure 3: Architecture weighting parameters by layer for one selected trial from rotMNIST, Galaxy10, and ISIC. For other trials, see Figures 8-10 in Appendix E.

results in Table 1 show that, the $C_1$-equivariant networks generally improve with either equivariance prior, while the $C_4$ equivariant networks perform better with $D_4$ equivariance initialization only when the $D_4$ constrained baselines also work well. The untrained prior methods do not perform as well as EquiNAS$_E$ on rotMNIST, showing the benefit of investing some training time to the constrained equivariance. For the other tasks, the baselines with priors have better performances than their constrained baseline counterparts.

### B.2. Differentiable Equivariance-aware NAS

The classification test errors are listed in Table 2. EquiNAS$_D$ achieves better test accuracy than the other comparable methods on rotMNIST and Galaxy10. Due to differences in training protocol, only comparisons of relative rankings with Table 1 are possible: baseline methods accuracies followed similar patterns to ranking between experimental paradigms, suggesting the benefit of general $C_4$ equivariance for rotMNIST and Galaxy10 and general

$D_4$ equivariance, including RPP, for ISIC. In this training protocol notably with adaptive optimizers, the results are more consistent across methods and trials.

The architecture weighting parameter dynamics for one exemplary trial per task are shown in Figure 3. The general trend of less constrained layers toward the end of the network supports the conjecture of local equivariance being beneficial. However, this effect is less consistent for ISIC with possibly less inherent symmetry: trials on ISIC, the only task where EquiNAS$_D$ did not exceed baselines, had the most varied architectures. As seen in Appendix E, the final mixing of architectures for ISIC included a high level of $C_1$, indicating that feature analysis outside of these symmetry groups is important for this benchmark.

Previous differentiable NAS works often used regularization of network size or even architecture weighting parameters themselves to encourage efficient architectures with a single highly weighted choice for each layer. However, our algorithm shows strong preference for a single, more equivariant and thus more expressive layer, notably to $D_4$ or $C_4$ equivariance, without such regularization. This may be due to the bilevel optimization dynamics: more constrained layers may be able to make more effective updates to more closely approximate the correct gradient computation that assumes optimal weights and thus become favorable compared to the lagging larger layers. This conjecture is shown particularly for trials on Galaxy10: increases in the architectural weighting parameter towards $C_1$ equivariance often comes after having strong weights for operations equivariant to larger groups.

### B.3. Discussion

To our knowledge, this is the first work which proposes search methods for networks with dynamically constrained equivariance. Many NAS approaches separately search for an architecture and then reinitialize and retrain the weights, while our two proposed approaches find an optimal architecture with trained weights in a single process, notably with dynamically constrained weights. Gradient-based tuning (Maclaurin et al., 2015) has shown the benefit not only of optimizing hyperparameters but also of dynamically adjusting them during training (Lichtarge et al., 2022). There is an inherent trade-off between accuracy and generalization capabilities with more constrained equivariance: dynamically constrained weights can reap both over the course of training.

Our two equivariance-aware NAS approaches have distinct approaches: EquiNAS$_E$ searches for architectures composed of discretely equivariant layers, while EquiNAS$_D$ searches for continuous mixtures of equivariance within each layer. The EquiNAS$_D$ algorithm avoids many known problems in differentiable NAS such as the discretization gap that occurs when searching over a continuous relaxation of a discrete architectural search space (Xie et al., 2021), such as that of EquiNAS$_E$. Towards searching for discretely equivariant layers using the $[G]$-mixed equivariant layer, proximal NAS algorithms use techniques such as projection (Yao et al., 2020) and straight-through estimation (Li et al., 2022) to avoid the discretization gap and thus may be effective for this application.

The EquiNAS$_E$ algorithm is innately greedy. At each selection step, the population is evaluated on the known current performance rather than the unknown final performance, so this metric is biased to architectures that train quickly. Networks constrained to higher symmetry group equivariance tend to learn faster, but this could be confounded by the equivariance relaxation causing large gradients for the newly unconstrained parameters and

thus potentially fast increases in performance. Further work could utilize other metrics for final performance, such as proxies (White et al., 2022).

All of the theoretical and algorithmic contributions of this work are applicable beyond the image classification experiments presented to architectures with parametrized equivariance to any discrete group. We leave the extension to other group representations and domains as future work.

Our proposed equivariance-aware NAS problems can be practically applied to find effective networks for datasets with hypothesizable symmetry. EquiNAS$_E$ may particularly work well on tasks that benefit from local equivariance, which can be determined by analyzing the architecture weighting parameters from first applying the more efficient EquiNAS$_D$. For tasks where EquiNAS$_D$ models have less consistent equivariance patterns, EquiNAS$_E$ could be adapted to propose candidates that relax any layer in the network, removing our constraint of non-increasing equivariance. We thus recommend EquiNAS$_D$ for practical applications if the final model is not restricted to discrete equivariance, in which case it can be used to inform design decisions for applying EquiNAS$_E$.

Beyond NAS, the equivariance relaxation morphism could be used in other applications such as fine-tuning and distillation. Layers of a pre-trained equivariant network could be expanded via equivariance relaxation before fine-tuning on the same or a new task. Similarly, a network could be distilled to a wider architecture for additional performance benefits.

The proposed equivariance-aware NAS algorithms are intentionally simple to focus on studying the architectural mechanisms presented in Section 2, although we have already discussed potential improvements to these algorithms. We include all valid results, even those that do not favor our own techniques, for transparency. This work aims to build a foundation for the intersection of equivariant architectures and NAS, rather than over-engineer the algorithms and experiments for incremental performance gains on selected benchmarks.

## Appendix C. Implementation Details

**Group convolutional layers**  The implementation of regular group convolutional layers can be viewed as a special case of our proposed equivariance relaxation morphism. With the preliminaries given in Section 2.2 and the case of $G' = T(2)$, $\tilde{f}$ and $\tilde{\psi}$ are computed such that $\tilde{f}_{(c,s)}(g') \coloneqq f_c(g's)$ and $\tilde{\psi}_{(d,t),(c,s)}(g') \coloneqq \psi_{d,c}(t^{-1}g's)$ for each $g' \in T(2)$, $c \in [C_{l-1}]$, $s, t \in R$, and $d \in [C_l]$.

Let $S_G \coloneqq |R|$. The learnable parameters of the $G_l$-equivariant $l^{\text{th}}$ layer with $C_l$ output channels, corresponding to $\psi$, are stored as a tensor of size $C_l \times C_{l-1} \times S_{G_l} \times K_l \times K_l$. The filter transformation expands this filter tensor by performing the action of each $r \in R$ on another copy of the tensor to expand its shape along a new dimension, resulting in a tensor of size $C_l \times S_{G_l} \times C_{l-1} \times S_{G_l} \times K_l \times K_l$, which is reshaped to $C_l S_{G_l} \times C_{l-1} S_{G_l} \times K_l \times K_l$. The input tensor to the $l^{\text{th}}$ layer, corresponding to $f$, is in the shape of $B \times C_{l-1} \times S_{G_l} \times H_{l-1} \times W_{l-1}$, which is reshaped to $B \times C_{l-1} S_{G_l} \times H_{l-1} \times W_{l-1}$ and convolved with the expanded filter. The output of shape $B \times C_l S_{G_l} \times H_l \times W_l$ is reshaped to $B \times C_l \times S_{G_l} \times H_l \times W_l$.

**Equivariance relaxation morphism**  To implement the equivariance relaxation morphism, the new filter tensor is initialized by applying Equation 4 such that result of applying the preceding filter transformation is equivalent. Our implementation of group actions

relies on group channel indexing to represent the order of group elements: to ensure this is consistent before and after the morphism, the appropriate reordering of the output and input channels of the expanded filter are applied upon expansion. The new filter tensor has a shape of $C_l|R| \times C_{l-1}|R| \times S_{G_l}/|R| \times K_l \times K_l$. The $[G]$-mixed equivariant layer is built on top of this implementation, also using proper input and output channel reordering between layers to ensure correct mixing of group channels.

## Appendix D. Experimental Details

**Architecture backbone**  For both EquiNAS$_E$ and EquiNAS$_D$ experiments, we use the same backbone architecture, such that the static baselines have the same architecture across experiments. The architectures have a lifting layer followed by 7 group convolutional layers, for a total of 8 convolutional layers. After 4 layers, the channel count doubles, from 16 to 32 for a $D_4$ equivariant layer and scaling up for smaller symmetry group equivariance constraints. An average pooling layer is placed after every other layer for all architectures and additionally after the fifth and seventh convolutional layers for Galaxy10 and ISIC. After the final group convolutional layer is a group-dimension average pooling followed by two linear layers to the output dimension. Every convolutional and linear layer except the output layer is immediately followed by a batchnorm then a ReLU.

**Hyperparameters**  The hyperparameters for each algorithm are selected such that baselines only differ by training time and optimizers. The learning rates were selected by grid search over baselines on rotMNIST. For all experiments in Section B.1, we use a simple SGD optimizer with learning rate 0.1 to avoid confounding effects such as momentum during the morphism. For EquiNAS$_E$, the parent selection size is 5, the training time per generation is 0.5 epochs, and the number of generations is 50 for all tasks. Baselines were trained for the equivalent number of epochs. For all experiments in Section B.2, we use separate Adam optimizers for $\Psi$ and $Z$, each with a learning rate of 0.01 and otherwise default settings. The total training time is 100 epochs for rotMNIST and 50 epochs for Galaxy10 and ISIC. For RPP, we use a $C_1$-equivariant layer with an $L2$ regularization parameter of $1 \times 10^{-6}$ in parallel with a $D_4$-equivariant layer without regularization.

For rotMNIST, we use the standard training and test split with a batch size of 64, reserving 10% of the training data as the validation set. For Galaxy10, we set aside 10% of the dataset as the test set, reserving 10% of the remaining training data as the validation set. For ISIC, we set aside 10% of the available training dataset as the test set, reserving 10% of the remaining training data as the validation. For the latter two datasets, we resize the images to $64 \times 64$ due to computational constraints and use a batchsize of 32. The validation sets were previously used for hyperparameter tuning: for experimental results, they are only used for the experiments in Section B.1 as necessary for the EquiNAS$_E$ algorithm. No data augmentation is performed, although the datasets are normalized.

## Appendix E. Additional Figures

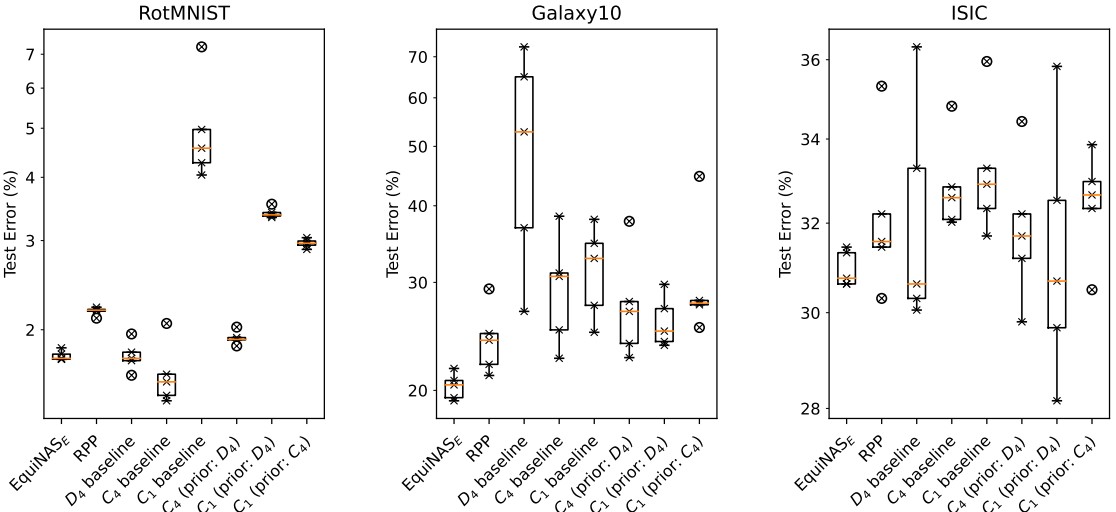

Figure 4: Test errors for experiments of Section B.1.

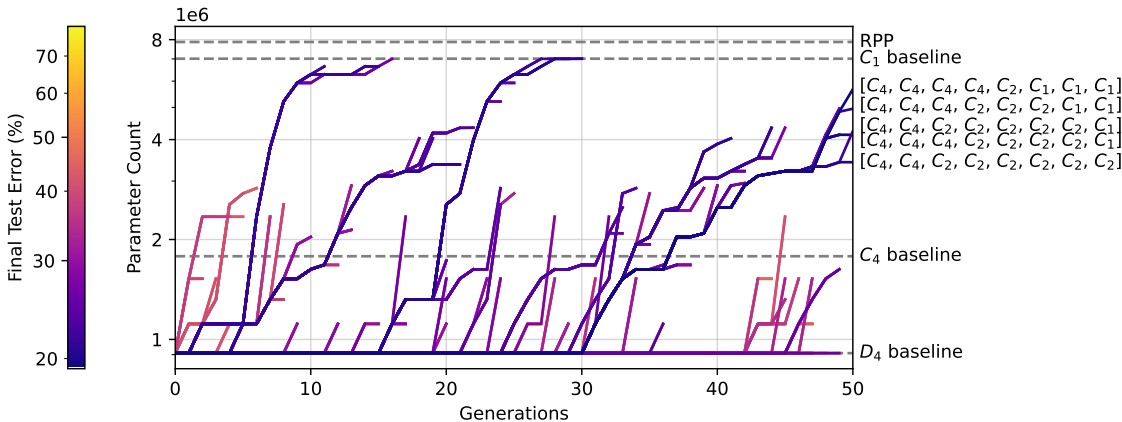

Figure 5: Historical parameter counts of each selected individual for EquiNAS$_E$ on Galaxy10.

Extended Abstract Track

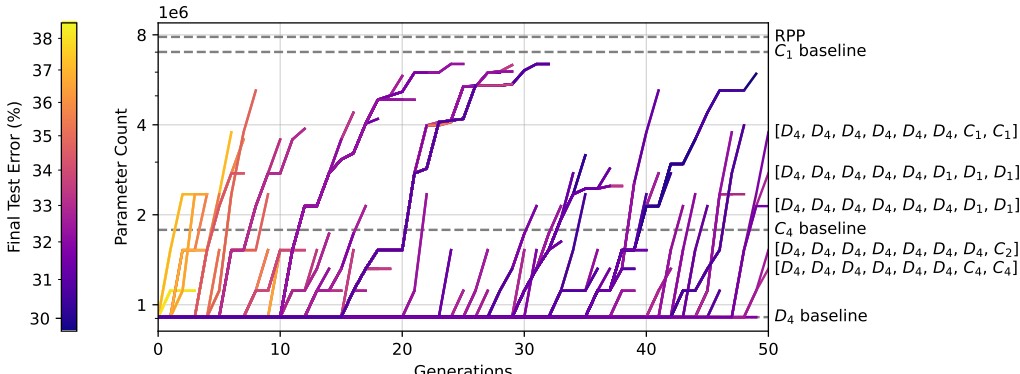

Figure 6: Historical parameter counts of each selected individual for EquiNAS$_E$ on ISIC.

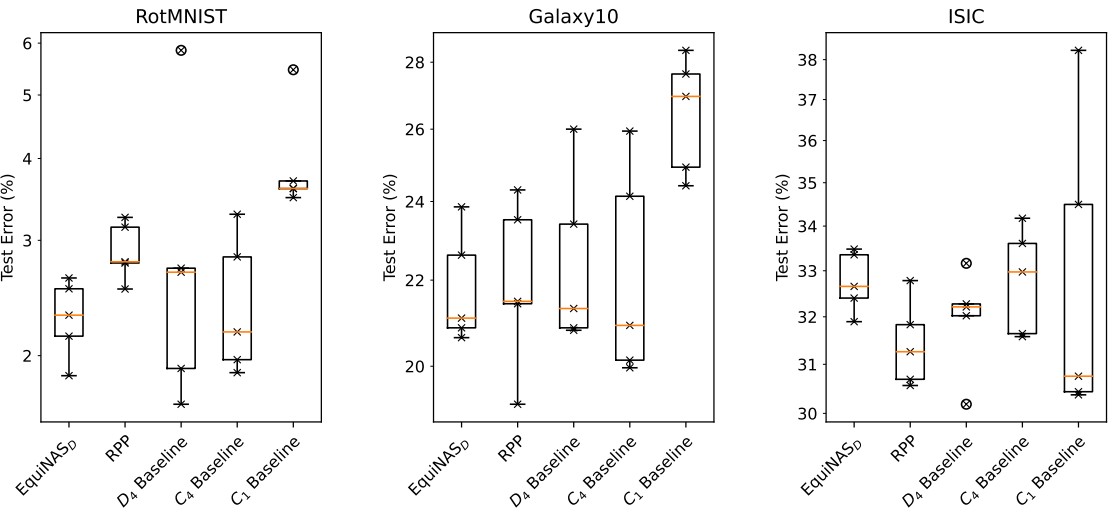

Figure 7: Test errors for experiments of Section B.2.

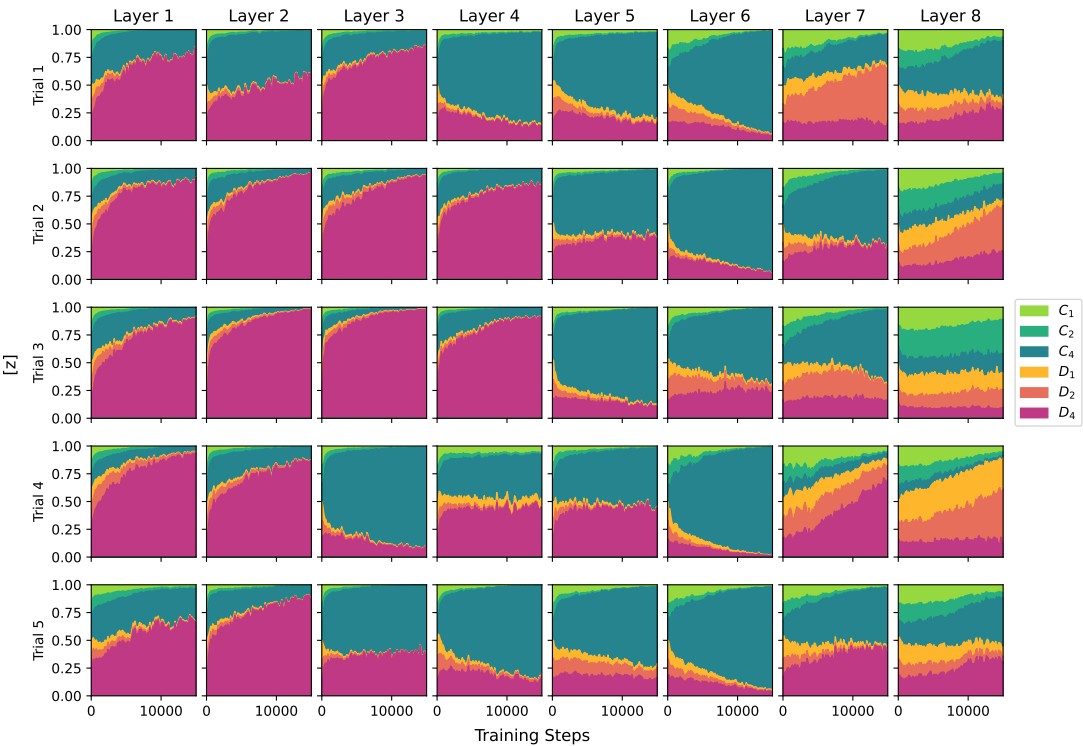

Figure 8: Architecture weighting parameters by layer for all trials on rotMNIST.

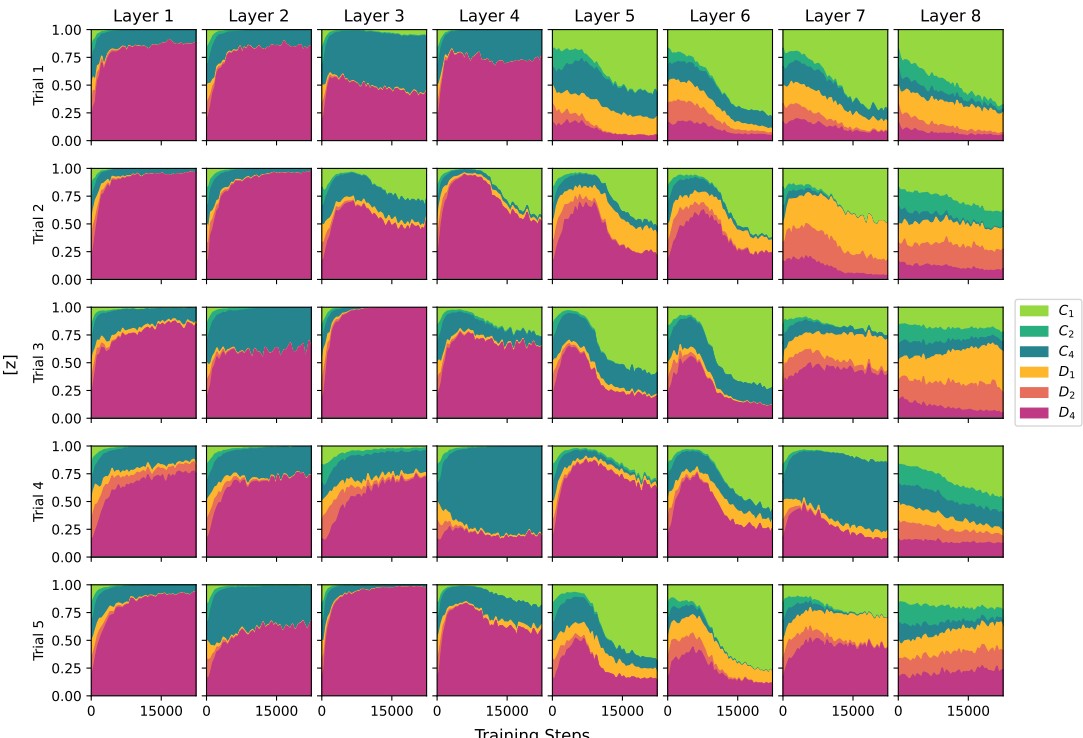

Figure 9: Architecture weighting parameters by layer for all trials on Galaxy10.

Extended Abstract Track

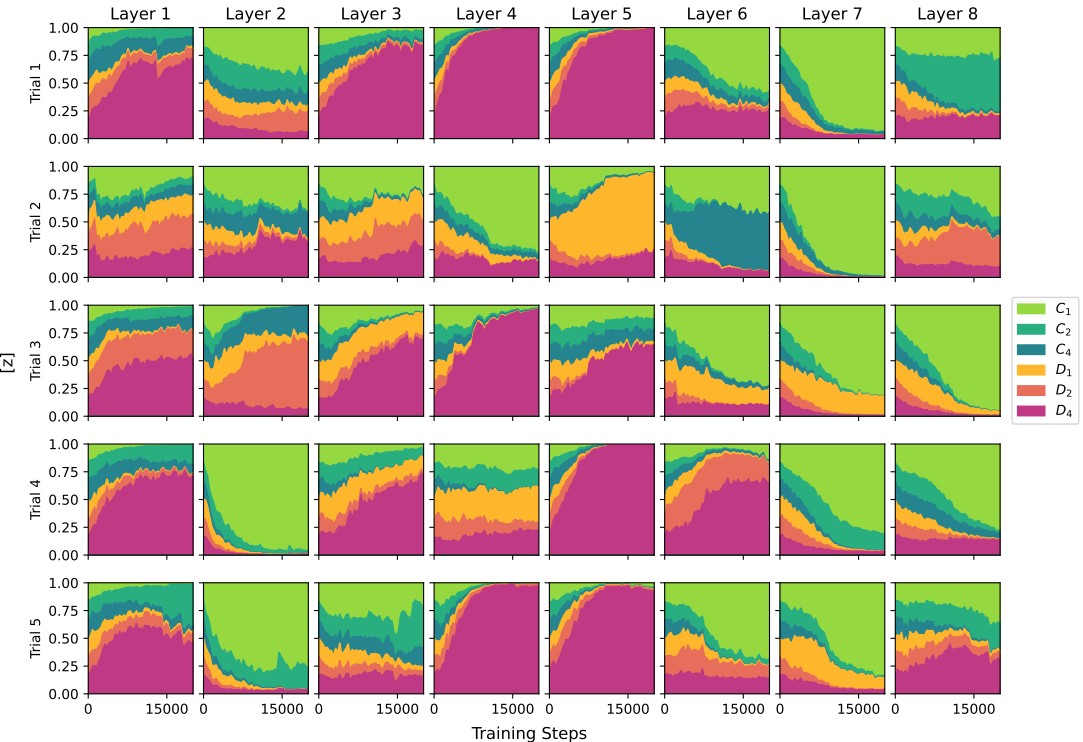

Figure 10: Architecture weighting parameters by layer for all trials on ISIC.

