# OpenReview forum: "Towards Architectural Optimization of Equivariant Neural Networks over Subgroups"
_NeurIPS.cc/2022/Workshop/NeurReps — NeurReps 2022 Poster_

### Official Review · Reviewer_EJKj · 2022-10-16
**Interesting extended abstract, but might be too early for workhop**

**Confidence:** 3
**Soundness:** 3
**Presentation:** 3
**Contribution:** 3
**Overall Rating:** 5

**Summary:**

This paper is an early-stage project.
The authors consider the problem of enforcing  equivariance in the layers of a deep neural network. Preserving symmetries has indeed been shown to be beneficial for algorithm transferability and performance. An important type of symmetries is equivariance: the output of a neural network applied to an image and its rotated counterpart should be identical. Currently, more work has been put into learning approximate equivariance.
Here the authors propose in extending this body of work via either (a) an equivariance relaxation morphism for group convolutional layers, and (b) a mixed equivariant layer that parameterizes a layer as a weighted sum of layers equivariant to different groups, so that it can represent equivariance to multiple subgroups through a weighted sum.


__Contributions:__
The authors introduce two new methods of enforcing equivariance: (a) an equivariance relaxation morphism that can be used jointly with an evolutionary NAS algorithm, and (b) a [G]-mixed equivariant layer that can be optimized with a differentiable NAS algorithm.


**Questions:**

See weaknesses

**Limitations:**

The authors do not discuss neither the context in which looking at subgroups rather than groups, nor the limits of this setting.

**Recommended Decision:**

2: Borderline

**Relevance:**

4: Highly relevant

**Strengths And Weaknesses:**

__Strengths:__
The approach seems novel.

__Weaknesses:__
This work is too preliminary.
 + First of all, the paper is perhaps a little unclear as to its primary objective. Two methods are presented, but they are not put enough in context (why are they useful, why are the authors suggesting two instead of just the one --- one of them is for NAS, and the other can be optimized over, sure.).
+ I did not understand very clearly the case for subgroups rather than groups --- this extended abstract would gain in clearly defining the end goal of the project and its intended purpose.
+ While the authors present a new type of convolutions, they do not have any type of experiment to back up their claims --- nor do they  try and explain how they would implement their method (and its computational cost). While the method is potentially interesting from a theoretical perspective, this work might unfortunately be too early stage to be part of this workshop.


**Submission Track:**

Extended Abstract (4 Page)

---

> ### Author Response · Authors · 2022-11-01
> **Response to EJKj**
>
> Thank you for your review. We hope our updated paper adequately addresses your concerns.
>
> We present these two methods because, although one somewhat builds off the other, they are useful in different contexts, both NAS and beyond, as shown by our newly proposed algorithms and experimental results.
>
> There may be a misunderstanding of our use of the term "subgroup". The equivariance relaxation morphism cannot be applied to generate a layer that is parametrized to be equivariant to just any group: it must be a subgroup of the group the layer is parametrized to be equivariant to.

---

### Official Review · Reviewer_YFhg · 2022-10-17
**Solid start to an interesting research direction. Hope to see these ideas explored in further depth.**

**Confidence:** 5
**Soundness:** 4
**Presentation:** 4
**Contribution:** 4
**Overall Rating:** 8

**Summary:**

"Towards architectural optimization of equivariant neural networks over subgroups"
This manuscript makes two proposals. First, (equivariance relaxation) given a fully equivariant convolutional network layer, they describe a procedure to prepare a different network  which computes exactly the same function but whose structure is no longer constrained to full equivariance over the original group and instead maintains equivariance to just a subgroup. Second, they propose to learn weights for partially equivariant networks (to multiple subgroups) in parallel as part of one network architecture, using pooling with a sum of representations

**Questions:**

Can the equivariance-relaxation be run in reverse? For example, given an MLP (or even just 1 fully-connected layer) that just happens to be equivariant, can you map it back to an equivariant group convolution? Further, if the network is not fully equivariant is it possible to 'compile' it into an equivariant one hurting the performance too much?

In some problems (e.g. HSP hidden subgroup problem) the group $G$ is known, and the subgroup $H\subset G$ is unknown. The goal is to find (e.g. by oracle access to some function on $G$) the generators of the subgroup $H$. I am just wondering are there any applications of your techniques to such problems?

Can we decompose a convolutional kernel into the components which are equivariant to the various points in the subgroup lattice of a group $G$? This could give some interesting plots :) For example it would be interesting to train a fully-equivariant convnet on some problem (perhaps affine MNIST), and then to lift (or relax) this constraint using the techniques in the paper, and to see how the equivariance respecting the various subgroups evolves. Maybe for some subgroups, the magnitude of the non-equivariant portion stays small, and for others it gets big. It would also be cool to measure the interplay with overfitting.

**Limitations:**

The techniques proposed by the authors are interesting and deserve further study. I have a few minor comments for small suggested changes below. I think the title and characterization of the work as "towards" optimizing over subgroups is an accurate portrayal of the work as it does not fully solve the problem in general (in general it is likely a very hard problem indeed...) but makes good suggestions for a few key problems.



One limitation of the abstract is that it does not contain specific proposals for experiments that could be run to quantify the performance of these methods. It is more of a theoretical paper. Another limitation is just that they are not clear which groups each technique is applicable to (I think it is just finite groups). These aspects might be clarified if edits are made to the manuscript.  I also suggested an experiment in the

**Recommended Decision:**

3: Accept

**Relevance:**

4: Highly relevant

**Strengths And Weaknesses:**

The idea of optimizing over subgroups is a natural problem. I think the authors contributions seem to pertain only to finite groups. I think for finite groups the relaxation technique is clear. However I’m not sure what happens when the system of representatives is infinite, which I think could be the case for eg a subgroup of a Lie group. It would be nice to see the authors clarify the applicability of their proposals to different groups. Perhaps the Lie group case can be handled with some small modifications to their recipe.

In fact the Lie group case is really interesting since in that case you could often write down the subgroup generators as continuous parameters (matrices) that you could learn directly with gradient descent.

The idea of interconverting between equivariant and non-equivariant models is fascinating and the authors give a clear description for how this could be done in the specific case of converting from a fully-equivariant group convolutional kernel (for a finite group) to a standard convolution. While these are only for specific cases, reading this gave me a lot of ideas and it would be a nice addition to the conference.

**Submission Track:**

Extended Abstract (4 Page)

---

> ### Author Response · Authors · 2022-11-01
> **Response to YFhg**
>
> Thank you for your thorough review. We believe our updates for the camera-ready version have addressed your noted limitations (at least those in-scope for an extended abstract), although it seems part of your text under "Limitations" was cut off, but perhaps it was referring to the decomposition experiment?
>
> Our methods are defined (and quite intuitive) for finite groups. However, we are definitely interested in extensions towards equivariance to groups with an infinite system of representatives as they are much more powerful. We leave this as future work.
>
> Regarding the "reverse" of the equivariance relaxation morphism, we investigated a mechanism for “projecting” a layer equivariant to some group G to one that is equivariant to a supergroup of G, so far only as a means of measuring the “distance” of a layer to equivariance to the super group, although this is not yet developed enough to be included in this work. Wang et al. 2022 use regularization of a similar distance metric as a comparison to their main proposed method. The projection could also be used towards "decomposing" a kernel into sub-kernels each with different equivariance properties, as you suggested. We also considered regularizing the operations of the [G]-mixed layer towards this idea.

---

### Official Review · Reviewer_qBVk · 2022-10-18
**A nice theoretical proposal for 2 techniques to search for sub-group equivariance via discrete/differentiable Neural Architecture Search. If supplemented with an empirical analysis of their use in practice, this will provide a solid contribution.**

**Confidence:** 4
**Soundness:** 3
**Presentation:** 2
**Contribution:** 3
**Overall Rating:** 5

**Summary:**

This paper proposes two *layer-wise* procedures (termed (1) equivariance relaxation morphism - for discrete NAS and (2) [G]-mixed equivariant layer - for differentiable NAS) to enable neural architecture *search over subgroups* of a given finite symmetry group G.

[C1] "Equivariance relaxation morphism": changes the equivariance constraint from one subgroup to another  subgroup while preserving the learned weights. This is achieved by reparameterising to partially remove weight-sharing constraints. It is mean for use in discrete architecture search algorithms.
[C2] "[G]-mixed equivariant layer":  represents equivariance to multiple subgroups (which are assumed to be totally ordered by inclusion) as a weighted sum, the weights of which can be optimised in a differentiable architecture search.

**Questions:**

[Q1] It would be great if the authors could provide an analysis of how the proposed methods work for NAS in practice. Since this is a theory paper, even a synthetic example would be enough to establish the usefulness of the propoced methods in combination with discrete/differentiable NAS algorithms.



**Limitations:**

There is currently no discussion of the limitations of the proposed methods.

In particular, I would be curious about the optimisation behaviour of the 2nd proposed method ([G]-mixed-equivariant layer) in an actual NAS setup. Since multiple subgroups are weighted differently, it seems there are "degenerate" weight combinations could lead to local minima that would need to be navigated around via the optimisation procedure/loss.


**Recommended Decision:**

3: Accept

**Relevance:**

3: Solid fit

**Strengths And Weaknesses:**

[S1] Motivation and relevance are well explained and contextualised at the beginning.
[S2] The paper is clearly written and provides technically sound justifications for the definitions.

[W1] While the layers are proposed and theoretically sound, there is no analysis on their behaviour and effectiveness in an actual NAS setup. This makes it hard to assess the practical significance of the proposed layers.
[W2] An visual illustration of each of the two procedures would strengthen the impact of this paper and make it more accessible to a larger audience. While it is understandable that the authors use mathematical language to state and prove their results, I would encourage them to include an intuition with a fitting figure to guide the reader.

*Disclaimer:* I am not an expert in the field of NAS, so I cannot comment on the originality of the work.

**Submission Track:**

Extended Abstract (4 Page)

---

> ### Author Response · Authors · 2022-11-01
> **Response to qBVk**
>
> Thank you for your review. We hope our updated paper adequately addresses your concerns, particularly adding visualizations of the two proposed mechanisms as well as two NAS algorithms with experimental results to address all noted weaknesses and questions.
>
> Regarding your point in "Limitations", this is something that differentiable NAS algorithms in general are conjectured to be prone to. We are considering regularization of each operation to ensure they not functionally equivariant to "higher" groups than their parametrized invariance for future work, although as with a lot of work in differentiable NAS, whether this choice would empirically improve the resulting architectures isn't clear.

---

### Author Response · Authors · 2022-11-01
**General response to reviews**

We would like to thank all three reviewers for their in-depth and constructive reviews. Our camera-ready version has been updated to address all in-scope concerns, particularly adding a visualization of each mechanism in the main text as well as proposing NAS algorithms for each mechanism with thorough experimental results in the appendices.

---

### Decision · Program_Chairs · 2022-10-21

Accept (Poster)